# Decoding the Reproductive System of the Olive Fruit Fly, *Bactrocera oleae*

**DOI:** 10.3390/genes12030355

**Published:** 2021-02-28

**Authors:** Maria-Eleni Gregoriou, Martin Reczko, Evdoxia G. Kakani, Konstantina T. Tsoumani, Kostas D. Mathiopoulos

**Affiliations:** 1Department of Biochemistry and Biotechnology, University of Thessaly, 41500 Larissa, Greece; magrigoriou@bio.uth.gr (M.-E.G.); kotsouma@bio.uth.gr (K.T.T.); 2Institute for Fundamental Biomedical Science, Biomedical Sciences Research Centre “Alexander Fleming”, 16672 Vari, Greece; reczko@fleming.gr; 3Department of Immunology and Infectious Diseases, Harvard T.H. Chan School of Public Health, 665 Huntington Avenue, Building 1, Room 103, Boston, MA 02115, USA; ekakani@gmail.com; 4Verily Life Sciences, South San Francisco, CA 94080, USA

**Keywords:** *Bactrocera oleae*, reproduction, oviposition rate, RNAi

## Abstract

In most diploid organisms, mating is a prerequisite for reproduction and, thus, critical to the maintenance of their population and the perpetuation of the species. Besides the importance of understanding the fundamentals of reproduction, targeting the reproductive success of a pest insect is also a promising method for its control, as a possible manipulation of the reproductive system could affect its destructive activity. Here, we used an integrated approach for the elucidation of the reproductive system and mating procedures of the olive fruit fly, *Bactrocera oleae*. Initially, we performed a RNAseq analysis in reproductive tissues of virgin and mated insects. A comparison of the transcriptomes resulted in the identification of genes that are differentially expressed after mating. Functional annotation of the genes showed an alteration in the metabolic, catalytic, and cellular processes after mating. Moreover, a functional analysis through RNAi silencing of two differentially expressed genes, *yellow-g* and *troponin C*, resulted in a significantly reduced oviposition rate. This study provided a foundation for future investigations into the olive fruit fly’s reproductive biology to the development of new exploitable tools for its control.

## 1. Introduction

Among arthropods, insects are the most divergent and abundant group, equipped with high reproductive rates and numerous post-mating behavioral and physiological adaptations critical to the maintenance and growth of their populations. The role of the reproductive system in these processes is central, and numerous studies have dissected its functions throughout the years. For the same reasons, the reproductive system is also the direct or indirect target of practically all means of insect pest control. As survival of a species is linked to reproduction, studying the reproductive strategies of an insect pest species can help identify new exploitable targets for the development of control methods.

Many such studies have focused on mating processes in different insects, since disrupting mating would reduce progeny production. During mating, sperm and seminal fluids produced in testes and male accessory glands (MAGs) with the ejaculatory bulb, are delivered to the female reproductive system to induce post-mating behavior [1,2,3]. Male seminal fluid proteins have been characterized mainly as proteases, peptidases, serpins, and protease inhibitors [4,5,6]. Although the functional classes of these proteins are conserved across species, their genes rarely are. Genes expressed in the accessory glands, as has been shown in Drosophila and Anopheles species, exhibit rapid evolutionary changes and gene expansion because of their critical role in encoding products that underlie striking, fitness-related phenotypes [7,8]. Until today, male seminal fluid proteins have been identified in a number of insects, including different Drosophilidae [9,10,11,12,13,14]; the major disease vectors *Anopheles gambiae* [15], *Aedes aegypti* [16,17], *Aedes albopictus* [18], and in Tephritid fruit flies like *Ceratitis capitata* [19,20] and *Bactrocera cucurbitae* [21].

Female accessory glands also produce a secretory material that serves several functions. It acts as a lubricant for egg passage, as a protective oothecal cover, or as a glue to attach eggs to various substrates [22]. To date, female reproductive genes in virgin flies have been comprehensively studied in very few insects, like the sandfly *Phlebotomus papatasi* [23], the house fly *Musca domestica* [24], and the Mediterranean fruit fly *C. capitata* [25,26]. After mating, the presence of sperm and seminal fluid in female insects induces multiple physiological and behavioral changes, such as the repression of sexual receptivity to further mating [27,28,29,30], stimulation of egg-laying, stimulation of immune responses, and reduced longevity [19,31,32,33]. These post-mating responses have been addressed in genome-wide studies in species such as *Drosophila melanogaster* [34,35], the honeybee *Apis mellifera* [36,37,38], *C. capitata* [39], *An. gambiae* [40,41], and *Ae.* a*egypti* [42]. Such studies have revealed that post-mating responses differ between different insect species.

For the olive fruit fly, *Bactrocera oleae*, the major pest of olive cultivation, the morphology and ultrastructure of the male accessory glands with ejaculatory bulb tissues were analyzed by Marchini et al. [43]. Moreover, the first molecular analysis of the reproductive system was presented in 2014 by Sagri et al. [44], who presented the identification of sex differentiation genes, i.e., differentially expressed genes either in male testes or female accessory glands and spermathecae [45].

Here, we focused on the mechanisms that underlie the post-mating responses of the olive fruit fly. As mating behavior is a very complex phenotype that includes pre- and post-mating processes, the analyses of gene expression changes are particularly appropriate for identifying the key genes or networks involved in mating. Based on the above, we performed an extensive transcriptomic and genomic analysis of the reproductive tissues from virgin and single-mated insects, which resulted in the identification of differentially expressed genes. To investigate their contributions in mating, we analyzed the expression profiles of selected genes throughout sexual maturation for the male tissues and different timepoints after mating for the female reproductive tissues. Finally, we silenced two of these genes, the *yellow-g* and *troponin C*, using RNAi technology, and we observed a significant reduction in oviposition. This study presents an integrated approach for the exploration of a physiological system, starting from the discovery of genes involved in the system’s function and leading up to the understanding of the gene’s roles through knocking down their expression and observing the generated phenotype. Furthermore, the information derived from this study not only sheds light on fundamental questions of the olive fly’s reproductive biology but also provides new exploitable tools for its control.

## 2. Materials and Methods

### 2.1. Ethics Statement

For these experiments, no specific permissions were required. This study did not interfere with any endangered or protected species, as it was carried out on laboratory-reared olive flies.

### 2.2. Fly Culture and Stock

The laboratory strain of the olive fruit fly was part of the original stock from the Department of Biology, “Demokritos” Nuclear Research Centre, Athens, Greece and has been reared in our laboratory for over 20 years. The flies are reared at 25 °C with a 12-h light/12-h dark photoperiod and humidity of 65% in laboratory cages with dimensions 30 × 30 × 30 cm^3^ with wax cones inside for oviposition, as described by Economopoulos, 1967 [40].

### 2.3. Tissue Collection

#### 2.3.1. Tissue Collection from Virgin Flies

Male and female flies were immediately separated after hatching (Day 0) in standard laboratory rearing cages. The insects were maintained separately until Day 7, when dissections were performed. The olive fruit fly laboratory strain used for the experiments was sexually mature and could mate successfully at the selected day. The sexual maturation of the insects was determined by their ability to mate and give offspring. Olive fruit fly mating takes place during the last 3 to 4 h of the photophase [46]. For this reason, the necessary tissues were collected during that specific timeframe during the day. For the RNAseq experiment, tissues from two pools of 50 insects (biological duplicate) were collected.

#### 2.3.2. Tissues Collection from Mated Flies

Male and female flies were immediately separated after hatching (Day 0) in standard laboratory rearing cages. The insects were maintained separately until Day 7. On Day 7, virgin male and female flies were mixed and allowed to mate. When mating occurred, single pairs were isolated from the cage and allowed to complete mating. A mating was considered successful if it lasted for at least one hour [47]. When insects completed mating (i.e., they voluntarily separated from each other), they were kept in different cages for 12 h. In *D. melanogaster*, the highest post-mating gene expression occurred after 6 h [34], and in *C. capitata*, there was a general increase in the transcriptional activity after only three repeated matings [48]. As there was no such evidence for the reproductive tissues of *B. oleae*, two pieces of information guided our decision for the determination of the appropriate timepoint after mating to analyze. Firstly, male olive flies can remate at least 24 h after a previous mating [49] and, secondly, oviposition of the mated females also starts 24 h after mating.

For the validation of the differentially expressed genes in testes, RNA was extracted from three pools, each one containing 10 pairs of testes. To determine the expression profiles of selected genes: (1) RNA was extracted from 10 pairs of ejaculatory bulbs/MAGs from virgin males at Day 0 (first day of hatching) until Day 7 (sexually mature), and (2) RNA was extracted from three pools of 10 pairs of the lower reproductive tract from single mated females immediately after mating (0 h) and 3, 6, 9, 12, and 24 h after mating for the female tissues. The normalized relative expression for the timepoints: Day 7 and 12 h after mating (AM) for the ejaculatory bulbs/MAGs tissue was used also for the validation of the differentially expressed genes for the males. The normalized relative expression for the timepoints: Day 7 before mating (BM) and 12 h AM, for the lower reproductive tract tissue, was used also for the validation of the differentially expressed genes for the females.

### 2.4. RNA Extraction

The dissected material was immediately immersed in the TRIzol Reagent (Ambion-Invitrogen). The RNA isolation was performed based on the TRIzol Reagent following the manufacturer’s instructions, with slight modifications. An additional DNA removal step using the TURBO DNA-free Kit (Ambion-Invitrogen, Austin, TX, USA) was performed according to the manufacturer’s instructions. RNA integrity was assessed by 1% agarose gel electrophoresis.

Following extraction, the RNA was treated with 1.0 unit of DNase I (Ambion-Invitrogen, Austin, TX, USA) according to the manufacturer’s instructions. The total amount of DNA-free RNA obtained from each tissue was converted into cDNA using 300-ng Random hexamer primers (equimolar mix of N5A, N5G, N5C, and N5T); 200 units MMLV Reverse Transcriptase (NEB, Ipswich, MA, USA); 5× Reaction buffer; 40-mM dNTP mix; and 40 units RNase Inhibitor (NEB), according to the manufacturer’s instructions. Reverse transcription was conducted at 42 °C for 50 min and 70 °C for 15 min. The resulting cDNA was used in subsequent qPCR reactions.

### 2.5. qPCR Reactions

Specific primers to amplify genes identified by the transcriptomic analysis were designed using the Primer-BLAST (http://www.ncbi.nlm.nih.gov/tools/primer-blast (accessed date 26 January 2021)) (Appendix A). Normalized relative quantitation was used to analyze changes in the expression levels of the selected genes using a Real-Time PCR approach. Expression values were calculated relatively to the housekeeping genes. *Rpl19* and *actin 3* were used as reference genes in all male reproductive tissue reactions, while *GAPDH* and *actin 3* were used as reference genes in FAGs/spermathecae reactions [50].

The qRT-PCR conditions were polymerase activation at 50 °C for 2 min and DNA denaturation step at 95 °C for 4 min, followed by 50 cycles of denaturation at 95 °C for 10 s, annealing/extension and plate read at 55 °C for 20 s, and finally, a step of melting curve analysis at a gradual increase of temperature over the range 55 °C to 95 °C. In this step, the detection of one gene-specific peak and the absence of primer–dimer peaks were guaranteed. Each reaction was in a total volume of 15 μL, containing 5 μL from a 1:10 dilution of the cDNA template, 2-SYBR Select Master Mix (Applied Biosystem, Foster City, CA, USA), and 300 nM of each primer. The reactions were performed on a Bio-Rad Real-Time thermal cycler CFX96 (Bio-Rad, Hercules, CA, USA), and data were analyzed through the CFX Manager™ software. All qRT-PCRs were performed using three technical replicates and three biological replicates (see Section 2.3).

### 2.6. Next-Generation Sequencing

RNAs were assessed on the Agilent Bioanalyzer system using the Agilent RNA 6000 Nano Kit reagents and protocol (Agilent Technologies, Santa Clara, CA, USA). Approximately, 0.5–1 μg of total RNA was used for mRNA isolation using the Dynabeads^®^ mRNA DIRECT™ Micro Kit (Thermo Fisher Scientific). The isolated mRNA was digested with RNase III, purified, hybridized, and ligated to Ion Adaptors. The samples were further reverse-transcribed, barcoded, and amplified using the Ion Total RNA-Seq Kit v2 (Thermo Fisher Scientific, Waltham, MA, USA). The quality of the libraries was assessed on the Agilent Bioanalyzer system using the Agilent DNA High-sensitivity Kit reagents and protocols (Agilent Technologies). RNA sequencing was performed in pools of 4 samples. Templating and sequencing took place on an Ion torrent system and an Ion Proton™ system [51], respectively, on Ion Proton PI™ chips (Thermo Fisher Scientific) according to the commercially available protocols. Sequencing was performed at the Fleming Institute (Greece) using the Ion Torrent™ Ion Chef™ automated platform as single end reads (GSE166965, GEO database). The cDNA library obtained from the testes of mated male insects (M_TESTES) was sequenced by Illumina Hi-Seq 2000 using the Illumina TruSeq RNA Sequencing protocol at the Genome Quebec in Canada as paired-end 100-nt reads (PRJNA288990, SRA database) of one biological replicate. The accession numbers for the Illumina libraries are SRR8800827 for the tissue of testes from virgin insects and SRR8800830 for the tissue of testes from mated insects [52].

### 2.7. In Vitro Double-Stranded (ds)RNA Synthesis

Target templates for in vitro transcription were generated using specific primers with the respective recognition site for T7 RNA polymerase (see S5) designed by the ERNAi [53]. A green fluorescent protein (GFP) gene was used as a template to synthesize the respective dsRNA used as a nontarget control. The sequences were amplified through PCR reaction. The amplified products were visualized and retrieved after agarose gel electrophoresis. The products were verified by DNA sequencing. The products were used as templates for in vitro transcription using the MEGAscript RNAi kit (Ambion, Austin, TX, USA), according to the manufacturer’s instructions. The dsRNAs were diluted to 1 μg/ul in phosphate-buffered saline (PBS).

### 2.8. dsRNA Treatment: Oviposition Rate

Upon emergence (Day 0), flies were separated by sex. For the ds-*yellow* RNAi experiment, 50 virgin male insects on their first day of hatching were injected with 69-nl ds-*yellow* gene (1 μg/μL) diluted in ddH_2_O. After injections, male flies were mixed with virgin female flies of the same age and were allowed to mate (for details, see Section 2.3). For the ds-*troponin C* RNAi experiment, 50 female insects on their first day of hatching (Day 0) were injected with 69-nl ds-*troponin C* (1 μg/μL) diluted in ddH_2_O. After injections, female flies were mixed with virgin male flies of the same age and allowed to mate (see Section 2.6). For each experiment, another group of flies was injected with 69-nl ds-*gfp* (1 μg/μL) that was used as a control. Flies were injected at the metathoracic segment using the Nanojet II (Drummond, Broomall, PA, USA) and glass needles under a Leica stereoscope.

To determine the oviposition rate, mating experiments were performed on Day 7 of the insects (as is described in Section 2.3.2. Tissues collection from mated flies). Each mated female fly from the experiments was placed in a separate cage with food, water, and an oviposition cone. Each cone was washed with dH_2_O daily, and the collected eggs were counted under a stereomicroscope for a 12-day period to record the oviposition rate of the insect.

For the determination of the knockdown efficiency, RNA was extracted from the lower reproductive tract (FAGs/spermathecae and uterus) or ejaculatory bulb/MAGs. Three biological pools were used, consisted of tissues from 10 flies each. The RNA isolation, cDNA synthesis, and qRT- PCR were performed as described above.

### 2.9. Bioinformatics Analyses

The representative de novo assembly was obtained with the Trinity pipeline (version 2.4.0) [54]. Transcript abundances were estimated using RSEM (version 1.3.1) [55]. Differential expression was assessed with the edgeR (version 3.8.0) algorithm [56]. Functional annotation was performed using the BLAST2GO (v.2) tool [57].

### 2.10. Statistical Tests

One-way ANOVA followed by Dunnett’s multiple comparisons test was performed using GraphPad Prism version 8.00 for Windows, GraphPad Software, La Jolla, CA, USA, www.graphpad.com (accessed on 26 February 2021). For simple comparisons, the unpaired Student’s *t*-test was used to analyze the significant differences. Values were stated as mean ± standard deviation (SD), and a *p*-value of < 0.05 was considered statistically significant.

## 3. Results and Discussion

In order to analyze the transcriptomic changes of the olive fruit fly reproductive system between virgin and mated insects, we collected the following tissues: (1) testes from mated males (M_TESTES); (2) male accessory glands with ejaculatory bulb tissues from virgin (V_MALE) and mated (M_MALE) males; and (3) the lower female reproductive tract, comprising of the spermathecae, the uterus, and the female accessory glands from virgin (V_FEMALE) and mated (M_FEMALE) females.

### 3.1. Transcriptome Sequencing Assembly

We generated a single representative de novo assembly of *B. oleae* from a concatenation of the libraries obtained with the Illumina platform [54] using the Trinity pipeline [54]. After assembly, using the RSEM [55], we calculated the transcript and unigene level expression values for the five libraries (M_TESTES, V_MALE, V_FEMALE, M_MALE, and M_FEMALE) sequenced by the Ion Proton™ system. A minimal quality control review for the libraries is presented in Appendix A (Appendix A and Appendix A). The average contig length was 669.78 bases (bp), and a total of 203,690,146 bp were sequenced, corresponding to 255,077 unigenes. The number of transcripts we obtained from the libraries were 3697 transcripts from the male testes library (M_TESTES), 11,452 transcripts from the male accessory gland libraries (V_MALE and M_MALE), and 10,478 transcripts from the female libraries (V_FEMALE and M_FEMALE).

To identify the differentially expressed genes between virgin and mated insects, we used the edgeR algorithm [56] with a stringent cutoff (q-value < 0.05). Females and males showed a similar number of upregulated genes after mating, while males had three times higher downregulated genes than females. Specifically, in the reproductive tissues from the mated males (male accessory glands and ejaculatory bulb), 1607 genes were upregulated, and 384 genes were downregulated, while in the tissues from the mated females, 1705 genes were upregulated, and 120 genes were downregulated (Appendix A). Furthermore, when we compared mated male testes (M_TESTES) to virgin testes (described by Sagri et al., 2014 [44]) transcriptomes, only 106 genes were upregulated, and 344 genes were downregulated. In general, fewer differentially expressed genes were detected in testes compared to the other two reproductive tissues, as presented in volcano plots (Figure 1), showing a limited transcriptional activity in the testes. This is in agreement with the results from *D. melanogaster*, where it was shown that spermatozoa are generally metabolically quiescent and transcriptionally silent in adult insects [58,59]. Complete lists of the annotated genes are given in Supplementary Appendix A.

Next, we performed functional annotation of the top 100 overexpressed transcripts in the post-mating tissues based on the Gene Ontology (GO) categorization level II using the BLAST2GO tool [57] (Figure 2, Table 1). Regarding the testes transcriptome, the most abundant hits on the GO functional annotation were based on categorization level II, like those obtained from the *Bactrocera dorsalis* respective testes transcriptomes [60]. Regarding the classification of the molecular function (MF), the identified genes mostly fell in the main groups of “binding” and “catalytic activity”. These groups were also identified as the most abundant in similar studies in *B. dorsalis* [60] and *C. capitata* [39,48], two close relatives of the olive fruit fly, showing conservation of the functions that alter during mating in these insects. The GO terms of the male accessory glands and ejaculatory bulb tissues, in response to biological processes and the molecular function, showed an enrichment of “metabolic processes” and “biological regulation”, as in *C. capitata* [39]. As sexually mature males are actively involved in the pheromone response and female courting, males show a significant enrichment of these GO terms, indicating the high energy investment required in mating. Finally, the GO annotation of the upregulated genes in the lower female reproductive tract of *B. oleae* showed genes encoding proteases, protease inhibitors and genes related to immune response and energy metabolism that are is in accordance with the data reports in other species [34,35,48,60,61,62]. This transcriptional activity of mated female olive fruit flies is characterized by rapid cell proliferation and secretory activity, as supported by the categorization of the transcripts in functional classes related to biological regulation, metabolic and cellular processes. However, a more detailed analysis of the transcripts showed that there is diversity in the mating response among species. Specifically, compared to *C. capitata,* there were two distinct differences. Firstly, there was a significant increase in the number of transcripts in single mated *B. oleae* insects. This increase was detected only in three times mated *C. capitata* insects. Secondly, a transcriptional change of the immunity response of the reproductive tissues was observed in *B. oleae*, while no such change was observed in *C. capitata* [39]. Such a divergence of the reproductive genes has been shown in other species and is based on the important role that they play in ensuring successful mating and fertilization [63,64]. The comparison of *D. simulans* male accessory gland proteins with their orthologs in its close relative *D. melanogaster* demonstrated a rapid divergence of many of these reproductive genes [65].

For the validation of the differential expression observed in the RNAseq of the testes, we performed a qRT-PCR analysis for nine loci. In testes of mated flies, statistically significant overexpression in the tissues isolated from insects after mating was confirmed for *c5823*, *hemolectin*. Overexpression in the tissues before mating was confirmed for the *scribbler isoform J*. Overexpression was detected also for *c37552, mucin* and *cation transporter* but with no statistical significance. qRT-PCR did not confirm the expression profile of the genes *c15699* and *c52071* obtained from RNAseq, as the results were contrasting. Based on the RNAseq, *c15699* was overexpressed in the tissues after mating, while the qPCR results showed no significant difference between the virgin and mated male insects. With regards to *c52071,* based on the RNAseq, it was overexpressed in the virgin flies, while the qPCR results showed an overexpression after mating. The *c42518* gene showed very low expression (Figure 3).

### 3.2. Genomic Annotation of Reproductive Genes

Reproductive genes were annotated to the sequenced genome of the olive fruit fly (https://i5k.nal.usda.gov/Bactrocera_oleae (accessed date 26 January 2021)) (Appendix A) following two procedures. First, the top 100 highly differentially expressed reproductive genes in each tissue (testes, male accessory glands and female lower reproductive tract) were annotated to the genome. Second, the genome scaffolds were queried (tBLASTn, e-value < 10^−10^) using the amino acid sequences of the 139 characterized *D. melanogaster* seminal fluid proteins SFPs [10] to annotate the additional SFPs. Only 43 of the Drosophila genes had significant hits in the olive fruit fly genome. The homologous genes were grouped into 17 functional classes based on the categories defined for the *D. melanogaster* [11] and *C. capitata* [20] seminal fluid proteins (Appendix A).

The annotated genes obtained from both of these procedures encode proteins that belong to the conserved functional classes such as proteases and protease inhibitors, lipases, sperm-binding proteins and antioxidants [66].

### 3.3. Transcriptional Analysis of the Differentially Expressed Genes

To gain better insight into the transcriptional changes of the reproductive genes, a more detailed follow-up expression profile analysis was performed for the male accessory glands with ejaculatory bulb tissues and the female lower reproductive tract.

For the male accessory gland genes, we determined the expression profiles of the genes *timeless*, *c52416, c53574, brunelleschi, yellow-g and CG2254-like* from Day 0 (first day of insect hatching) to Day 7 (sexually mature insects) virgin insects. We selected these genes based on their differential expressions between virgin and mated flies. As it is presented in Figure 4, the comparison between gene expression at timepoint 7 (Day 7 virgin males) and 12 AM (12 h after one mating) indicates that the genes are significantly overexpressed after mating, confirming the RNAseq results. Our assumption is that a gene encoding a seminal fluid protein should be expressed before mating, so that the protein will be present at the time of mating. In agreement with this hypothesis, the highest expression of most genes was detected before Day 7 (Figure 4). Specifically, the genes *timeless*, *c52416* and *c53574* showed the highest expression on Day 0, while their expression dropped to lower levels and remained stable until Day 7. *Timeless*, along with the *per* (*period*), *Clk* (*clock*) and *cyc* (*cycle*) genes, regulate the circadian cycle of the insects. In most Bactrocera species, mating occurs at a species-specific time window during the day where male sexual activity should be synchronous with female receptivity [67], modulated by the circadian clock, as it is in Drosophila [68]. The knockout of *timeless* in male *D. melanogaster* resulted in a decreased reproductive fitness in males [69]. In *Spodoptera littoralis*, it has been demonstrated that the sperm release rhythm is controlled by an intrinsic circadian mechanism located in the reproductive system [70]. The presence of the *timeless* gene in the male reproductive tissues may indicate that there is a similar circadian mechanism in the olive fruit fly that controls important reproductive events, e.g., sperm release.

The other three genes analyzed from the male reproductive tissues were *brunelleschi, yellow-g* and *CG2254-like. Brunelleschi* and *yellow-g* showed their highest expression on Day 5 and gene *CG2254-like* on Day 6. *Brunelleschi* encodes a protein that belongs to the TRAPII complex, which is involved in vesicle trafficking in the secretory pathway [71]. As it was mentioned above, the male accessory glands are the secretory tissues of the reproductive system. Based on the gene expression profile, it may be involved in the maturation of accessory glands to produce the secretory proteins of the seminal fluid.

*Yellow-g* belongs to the MRJP/YELLOW family, which includes the major royal jelly proteins and the yellow proteins. The *yellow* gene family is associated with behavior [72,73,74,75,76,77], pigmentation [78,79] and sex-specific reproductive maturation [80] in *D. melanogaster* and *A. mellifera*. In 2019, Massey et al. showed that the *yellow* gene in Drosophila affects the male mating success through sex comb melanization. The loss of *yellow* expression in these modified bristles reduces their melanization, which changes their structure and causes difficulty in the male grasping of females prior to copulation [81]. The pleiotropic effects of *yellow* on male mating success might result from the effects of *yellow* in the adult central nervous system. The highest expression of the *yellow-g* gene on DAY-5 in our study indicates that it may encode a protein that plays a role in the maturation of the male reproductive tissues or in the male mating success, as its protein should be present when the male is ready to mate.

*CG2254-like* encodes a dehydrogenase that is localized in the lipid droplets, organelles that store lipids and has a significant role in metabolism and membrane synthesis [82]. In *D. melanogaster* adult stages, *CG2254* expression is fairly widespread and strong in tissues that have a lipid storage function, as well as in tissues that have not so far been shown to have a clear-cut lipid storage function, such as the eye, the brain and the spermatheca, suggesting a prominent role of the protein within bona fide, as well as nonclassical, lipid-storing tissues [83]. The presence of the gene in our findings may indicate a similar role in the olive fly. Specifically, the ejaculatory bulb is a muscle tissue, and its contractions help seminal fluid transfer to the female flies during mating. During mating, the tissue has high energy demands, and the presence of the CG2254-like protein may be an alternative energy source.

Regarding the female reproductive tract, the expression profile of the following genes was determined in Day 7-old virgin females and at six timepoints (0, 3, 6, 9, 12 and 24 h) after mating: *troponin C, ornithine decarboxylase antizyme, lingerer, bestrophin-2*, *yolk protein-2* and *glutathione S-transferase* (Figure 5). We selected these genes based on their differential expression between virgin and mated flies. As it is presented in Figure 5, timepoints 7D BM (Day 7 virgin females) and 12 (12 h after one mating) are the same timepoints we used for the RNAseq analysis. In four out of the six genes, *ornithine decarboxylase antizyme, lingerer, bestrophin-2* and *yolk protein-2*, the qRT-PCR results confirm the original findings of the RNAseq analysis. In two cases, though, *troponin C* and *Glutathione S-transferase*, the qRT-PCR results seem to contradict the RNAseq. While, in general, there is very good correlation between RNAseq results and qPCR data [84,85,86]), discrepancies may also arise. From a technical standpoint, RNAseq experiments assess the entire transcript in an unbiased manner [87], while in a qRT-PCR, only a small region of the cDNA is amplified (probe bias). In fact, it has been shown that 3′-UTRs are quite variable, and excluding them from the RNAseq data improves the consistency of the RNAseq and qRT-PCR results significantly [88]. From a biological standpoint, various types of small asynchronies in biochemical or molecular processes of the biological material could lead to differences in gene expression. For these reasons, RNAseq results are often validated by qRT-PCR, which, in addition, may be enriched with the analysis of diverse samples, capturing more details and more timepoints.

To determine the expression profile of the genes in the female lower reproductive tract, we hypothesized that if a gene codes for a protein that is induced because of mating, it should be expressed some time after mating. The obtained expression profiles of the six genes were variable. *Troponin C* showed limited expression after mating. *Ornithine decarboxylase antizyme* showed an increasing expression with the highest expression at 24 h after mating, while *lingerer* (10-fold) and *bestrophin-2* (10-fold) showed the highest expression at 12 h. *Yolk protein-2* showed two-fold overexpression nine h after mating, and *glutathione S-transferase* showed the highest expression immediately after mating (zero h).

*Troponin C* protein plays a significant role in muscle contractions. In *Pieris rapae*, the small cabbage white butterfly, it was identified as a component of the bursa copulatrix female reproductive tissue that is responsible for the digestion of the nutrient-rich spermatophore produced by the male accessory glands [89]. The overexpression of this gene in the female reproductive tract of virgin flies indicates its involvement in muscle contractions, probably aiding in the digestion of the seminal fluid proteins that are transferred to the female during mating.

An upregulation of Ornithine decarboxylase antizyme (ODC-AZ) was observed in mated females. ODC-AZ binds and destabilizes the ornithine decarboxylase (ODC), a key enzyme in polyamine synthesis [90]. Correlative changes between hormone levels and polyamine metabolism were described in several insects. For example, 20-hydroxyecdysone increases ODC activity in silk moth pupal tissues [91] and juvenile hormone stimulates ODC activity during vitellogenesis in *D. melanogaster* [92]. Ornithine decarboxylase antizyme is an inhibitor of ODC. The inhibition of ODC activity causes impaired vitellogenesis in *Ae. aegypti* [93] and oviposition delay in the silkmoth *Hyalophora cecropia* [91]. The observed upregulation of their inhibitor indicates that ODC-AZ is probably involved in the control of ODC levels in mated female olive fruit flies.

*Lingerer* showed an upregulation 12 h after mating. Mutations of *lingerer* in male *D. melanogaster* result in abnormal mating and the “stuck” phenotype, where males cannot be separated from females after the end of mating. It has also been identified as a maternal gene expressed in *D. melanogaster* early embryos. According to the literature, *lingerer* is expressed at high levels in the nervous system, imaginal discs and gonads and is required for sexual behavior and viability [94]. The overexpression of the gene 12 h after mating may indicate that it is transferred as maternal gene to the offspring.

A similar expression profile has been demonstrated for the *bestrophin 2* gene. In *D. melanogaster*, it encodes an oligomeric transmembrane protein that is thought to act as chloride channel [95]. The observed upregulation of the gene in mated female insects may indicate its role in the transportation of small molecules that are transferred into the female flies as part of the seminal fluid during mating.

The *Yolk protein-2* homolog in *D. melanogaster* is expressed almost exclusively in females, and it was associated with a female sterile mutation [96,97]. The *Yolk protein-2* gene encodes for a precursor of the major egg storage protein, the vitelline. Three main factors regulate vitellogenesis in *D. melanogaster*: a brain factor, an ovarian factor that stimulates fat body vitelline synthesis and a thoracic factor that is involved in the uptake of the vitelline by the ovaries [98]. Moreover, vitellogenins are implicated in the transportation of various molecules, such as sugars, lipids and hormones, in insects [99].

The Glutathione S-transferase epsilon class is a predicted intracellular or membrane-bound protein [100]. Predicted intracellular proteins have been reported in the reproductive system of *D. melanogaster* [4], *A. mellifera* [101] and *Ae. aegypti* [102]. For both *A. mellifera* and *Ae. Aegypti*, these proteins are suggested to be secreted through apocrine and holocrine secretion, nonstandard secretion routes [103,104]. Macro-apocrine secretion has been reported in the *B. oleae* male reproductive system [43].

### 3.4. Inhibition of Reproductive Genes through RNAi Silencing

We performed transient inhibition through RNAi silencing to test the functional role of the genes in the male and female reproductive systems. For the olive fruit fly, this method was successfully implemented through injections in embryos [105] and adults [106] and, also, through feeding in adult olive flies [107]. We selected the *yellow-g* and *troponin C* genes for the male and female reproductive systems, respectively. We selected these genes based on their logFC value from the RNAseq. Moreover, their putative roles, as described above, made them interesting candidates for functional analyses.

dsRNAs targeting the two genes were injected into the hemolymph of virgin adult male (ds-*yellow*) and female (ds- *troponin C*) insects on their first day of hatching (Day 0). Injection of the ds-*yellow* in male insects resulted in 46% and 81% downregulation of the gene on Day 5 and Day 7, respectively, compared to the control group (Figure 6A). Similarly, injection of the ds-*troponin C* in females caused 70% and 64% downregulation of the gene on Day 4 and Day 7, respectively, compared to the control (Figure 6C). The successful response of the fly to the RNAi process should not be taken for granted, as the RNAi response differs between different insects and different genes. In Drosophila, RNAi-mediated gene knockdown through microinjection was only localized to the site of dsRNA delivery, and the effects were temporally limited [108]. On the contrary, injection of dsRNA into the adult abdomen of *B. dorsalis* successfully inhibited the expression of *doublesex* in ovaries [109]. Our RNAi experiments demonstrated a systemic inhibition in the olive fruit fly more like what was observed in *B. dorsalis* rather than in *D. melanogaster*.

To determine whether these genes could affect the insect’s reproductive activity, we performed two different mating experiments and recorded the oviposition rates. We chose this trait, as we wanted to determine if there is a possible impact on the population size of the insects, the main purpose of the pest control methods.

Females mated with ds-*yellow*-injected males showed the same pattern of oviposition rate per day as females mated with ds-*gfp*-injected males (control): egg laying peaked on Day 3, followed by a regressive phase, with the lowest counts recorded on Day 12 (Figure 6B). However, the daily egg-laying rate of the females mated with ds-*yellow*-injected males was significantly reduced compared to the control females (*p* < 0.05). The reduced number of eggs may have been caused by the reduced male mating success of the insects, as reduced mating success could lead to fewer fertilized eggs and, therefore, a decreased oviposition rate. This is in agreement with the role of *yellow-g* in male mating success by sex comb melanization in *D. melanogaster* [81]. Until today, there was no information about the sex comb in the olive fly. However, the gene could play a similar role in a sex-specific tissue responsible for olive fly male mating success.

On the second mating experiment, ds-*troponin C* (ds-*TnC*)-injected females showed a significantly reduced daily oviposition rate from Day 6 after mating compared with ds-*gfp*-injected females (control group) (*p* < 0.05) (Figure 6D). As it was mentioned above, *troponin C* participates in muscle contractions. The reproductive tract is primarily an epithelium surrounded by circular muscles [110]. Muscular contractions are significant for the proper function of the reproductive system. Movements of the upper oviducts are responsible for the transportation of the eggs through the genital chamber, and contractions of the spermatheca lead to sperm release, resulting in the fertilization of eggs. Based on the above, if the muscle system in the reproductive tissues is not properly shaped, it would affect reproduction. This may explain why the silencing of *troponin C* led to a decreased oviposition rate.

## 4. Conclusions

The present study shows a comprehensive approach of how to explore one of the major systems of living organisms, the reproductive system. We started from a holistic transcriptomics analysis and managed to identify specific genes whose inhibition of expression gave interesting phenotypes, such as the reduction of the oviposition rate of female insects. Clearly, such reproductive genes, identified through these types of analyses, could be important targets for the development of insect control methods.

During the last sixty years, agrochemical companies have been in a constant quest for more specific, effective and environmentally friendly approaches to contain populations of innocuous insects. RNA interference has been explored as a strategy for pest control by administering insect-targeted double-stranded RNA to specifically block the expression of essential genes [105,111,112]. Obviously, reproductive genes are ideal targets in such approaches. Alternatively, as CRISPR technology has been implemented successfully in the olive fruit fly [113,114], one can envision a CRISPR-based gene drive system in which a Cas9 (with its guide RNA) targets an essential reproductive gene [115,116]. The result would be a rapid substitution of the gene in a population with an impaired copy (or the complete deletion of the gene), thus damaging the pest’s reproductive ability and leading to its population collapse.

## Figures and Tables

**Figure 1 genes-12-00355-f001:**
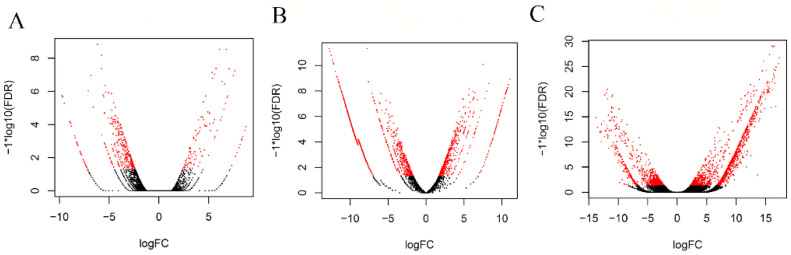
Volcano plots representing the differentially expressed genes between virgin and mated flies in the three dissected tissues: (**A**) testes, (**B**) male accessory glands with ejaculatory bulb and (**C**) lower female reproductive tract. The Y-axis represents significance, and the X-axis represents logarithmic fold change. The red dots represent differentially expressed genes with a *p*-value < 0.05.

**Figure 2 genes-12-00355-f002:**
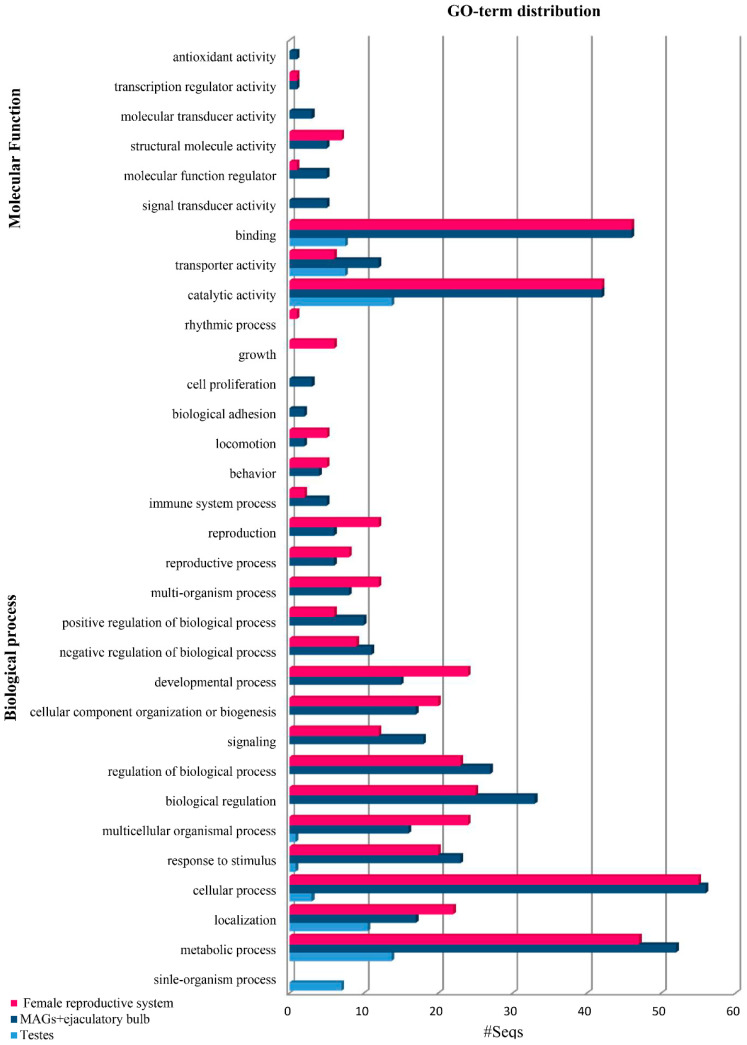
Gene Ontology (GO) terms (level 2) distribution of the top 100 overexpressed genes in *Bactrocera* oleae reproductive tissues from mated insects showing the top 20 hits of different categories for the molecular function (MF) and biological process (BP).

**Figure 3 genes-12-00355-f003:**
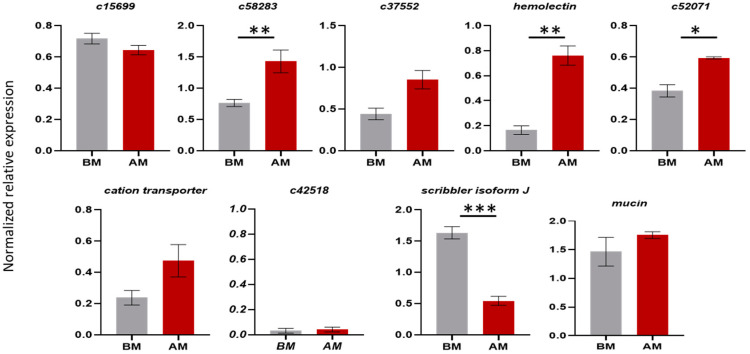
Comparison of the expression levels of genes expressed in male testes from Day 7 virgin insects before mating (BM) and testes from insects 12 h after one mating (AM). Mean values ± standard deviation (SD) of three biological replicates are shown. The * indicates significant differences, as determined by an unpaired Student’s *t*-test: *p*-value < 0.05 (*), *p* < 0.021 (**), *p* < 0.0002 (***).

**Figure 4 genes-12-00355-f004:**
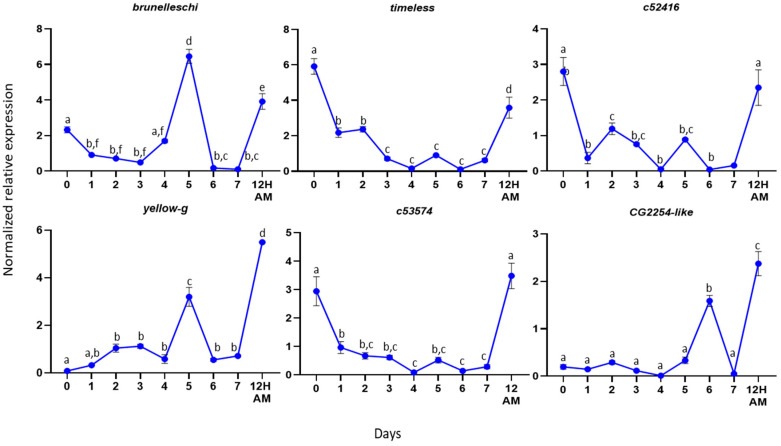
Expression profiles in male reproductive tissues. Expression profiles of the selected genes from the first day of hatching (Day 0) until Day 7. The error bars show the standard error of the mean between three biological samples. Small different letters next to the SD bars indicate statistically significant difference among different samples (*p* < 0.05, one-way ANOVA with Tukey’s test).

**Figure 5 genes-12-00355-f005:**
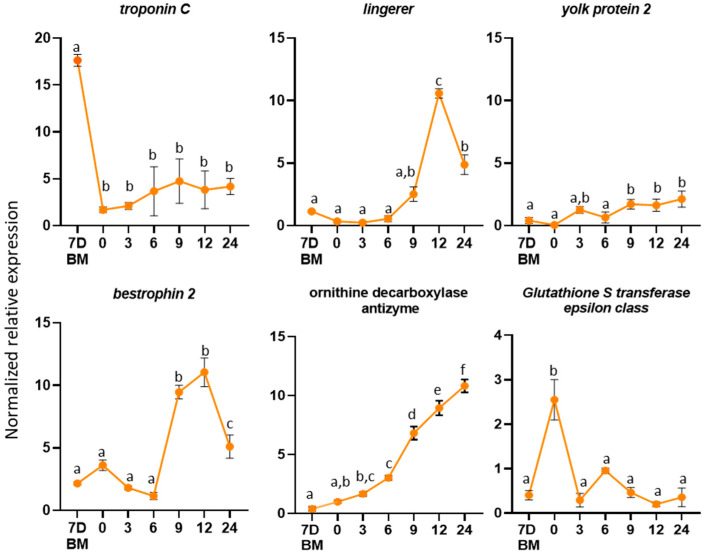
Expression profiles in female reproductive tissues. Expression profiles of the selected genes from the virgin flies and several timepoints after mating (0, 3, 6, 9, 12 and 24 h). The error bars show the standard error of the mean between three biological samples. Small different letters next to the SD bars indicate a significant difference among different samples (*p* < 0.05, one-way ANOVA with Tukey’s test).

**Figure 6 genes-12-00355-f006:**
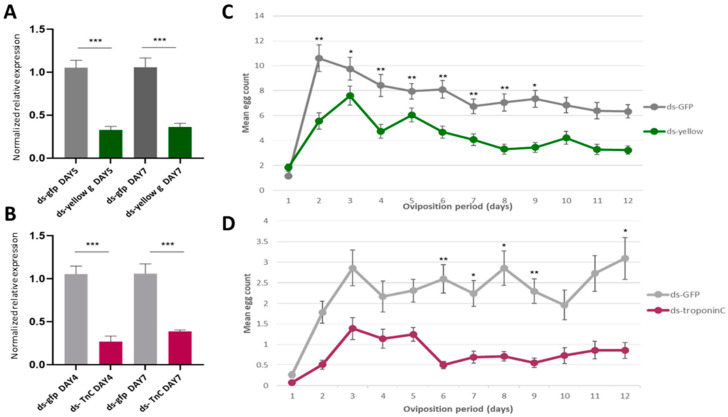
The effect of *yellow-g* and *troponin C* knockdown to oviposition. (**A**,**B**) Relative quantitative analysis of the *yellow-g* gene (**A**) and the *troponin C* gene (**B**) in the double-stranded (ds)RNA-injected *B. oleae* insects. Expression levels were evaluated relative to the control injections with ds-*gfp*. All values were normalized using *Rpl19* and *actin3* as the reference genes. Data are presented as means ± standard deviation (SD). Asterisks indicate the significance level of the difference between the two samples (*p* < 0.05, unpaired Student’s *t*-test). (**C**,**D**) Oviposition activity of treated females during a 12-day period. Day count starts one day after mating assay. Mean values ± standard error of triplicate data from three biological replicates is shown. (**C**) The mean daily egg count for the females mated with males injected with ds-*yellow* (green line) and males injected with ds-*gfp* (grey line). (**D**) The burgundy line shows the oviposition rate of the females injected with ds-*troponin C* and the grey line shows the oviposition of the control group (ds-*gfp*). The asterisks indicate statistically significant difference: *p* value < 0.05 (*), *p* < 0.021 (**), *p* < 0.0002 (***) (one-way ANOVA followed by Dunnett’s multiple comparisons test).

**Table 1 genes-12-00355-t001:** List of the differentially expressed genes of the RNAseq analysis selected for validation. The name used is based on their homologue in *D. melanogaster*. Genes that have no hits are presented with their transcript name. A positive value of logFC of testes, represents the overexpression of the genes in mated flies while negative value of logFC represents the overexpression of the genes in virgin flies. A negative value of logFC of MAGS/ejaculatory bulb or female reproductive tract indicates overexpression of the gene in mated flies while positive logFC value represents overexpression of the gene in virgin flies.

Name	LogFC	Name	LogFC	Name	LogFC
Testes		MAGs/Ejaculatory Bulb		Female Lower Reproductive Tract	
*c15699*	8.7	*timeless*	−11.7	*troponin C*	−14.9
*c58283*	8.6	*c52416*	−9.7	*yolk protein-2*	−14.6
*c37552*	7.5	*CG2254-like*	−12.3	*lingerer*	−11.7
*hemolectin*	7.6	*brunelleschi*	−9.5	*glutathione S-transferase e class*	8.8
*cation transporter*	6.6	*yellow-g*	−10.5	*bestrophin 2*	−15.4
*c42518*	5.5	*c53574*	−11.7	*ornithine decarboxylase antizyme*	−14.3
*scribbler isoform J*	−3.3				
*mucin*	8.2				
*c52071*	−4.8				

## Data Availability

Data is contained within the article or supplementary materials.

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
