# Peer review of "Decoding the Reproductive System of the Olive Fruit Fly, Bactrocera oleae"

_genes, 2021, doi:10.3390/genes12030355_

Round 1

Reviewer 1 Report

In this paper, Gregoriou et al. analyzed the transcriptome of reproductive tissues from virgin and mated individuals to better understand the mating mechanisms in the agricultural relevant insect Bactrocera oleae. After confirmation of the results, the authors selected two genes for RNAi silencing followed by a test of their potential function in the female reproductive system.

This study is interesting because it provides novel descriptive information useful for the field and an attempt to unravel potential mechanism through which two of these genes could act on female mating. However, this study suffers major flaws in its technical design, analysis, and interpretation that I will describe below.

Major comments:

  • RNAseq analysis:

The transcriptomic approach and its comparison with previously published data is interesting and could lead to important observations. However technical imprecision and an analysis too superficial make the results not reliable. First of all, I could not find any link to the raw sequencing data obtained in this research. Deposition of the sequencing data in a database (such as geo on the ncbi website) is mandatory for publication of transcriptomic data.

The Methods section does not provide enough information on how the reads were sequenced (length, single or paired-end, …) and no information at all on the bioinformatics analysis. For example how the data were aligned? Which software (and version) were used?

A minimal quality control should be presented to the reader to judge the quality of the experiment and therefore its relevance. I suggest that at least basic statistic on reads (number, % aligned…), correlation between all samples, and principal component analysis should be provided in order to assess the reproducibility of the data. Also please add the published data on virgin testes from Sagri et al. (2014) to this analysis. As it was generated for a different publication in a different context, one should make clear that the differential expressed genes are not due to technical differences but are biologically relevant.

  • RT-qPCR:

RT-qPCR experiments were used with three purposes: confirm RNAseq data, perform a time course analysis on selected genes, and confirm RNAi efficiency.

In all cases, only one reference gene was used per tissue type. Best practice in qPCR suggests that, at least, 2 references genes should be used in order to obtain trustworthy results. The authors should then add at least one extra reference gene for all their analyses.

For many qPCR based figures, there is no statistical analysis at all (Fig.4, 5) or the type of test is not mentioned (Fig6 A,B). Without statistical analysis, no conclusion can be drawn from these figures. Also, for clarity, the type of test used should be mentioned both in Materials and Methods section and on the legend of the figures.

Figure 2 and 3 do not confirm RNAseq results very well, only few genes show statistically significant differences in mated and virgin conditions and one gene is even behaving in the opposite direction. Using additional reference genes could improve the qPCR results. If RNAseq and RT-qPCR lead to different conclusions, then it should be discussed more in details.

Figure 3 appears redundant with figure 4 and 5 and could be removed. If I am not mistaking, comparison between “day 7” and “mated” in Fig.4 is identical to results in Fig.3A, while comparison between “virgin” and “12 hours mated” in Fig.5 is identical to Fig.3B. Then, if Figure 3A is identical to figure 4, why are the error bars different?

Finally I would like to add that figures 2 and 3 are conceptually wrong. RT-qPCR only provides semi-quantitative results. Showing different genes on the same graph suggests that the measures are absolute and the level of gene X could be directly compared with gene Y, which is not the case. Every gene should therefore be presented on a different graph.

  • Oviposition rate:

The functional analysis on two genes identified with RNAi is an interesting attempt to explain the potential function of these genes. However the data remain preliminary and the observed variable questionable.

Indeed, I am not convinced that looking at female egg count when RNAi was carried out in males is fully relevant. The hypothesis of the authors is that yellow g may be important for males to achieve proper mating with the females. So why not looking at mating success, for example by looking at how long is the mating process, how many attempts need the male to mate...

The materials and methods for this section is not clear. It reads “males were mixed with virgin female flies of the same age and were allowed to mate” (lines 196-197). Does this mean that all flies were allowed to mate as a batch, or were individual crosses done? In case of batch mating, a male with a higher fitness would be able to mate several times, introducing a bias in the results.

In both experiments (although not significant in D), the RNAi is affecting the egg count as early as the 2nd oviposition day. However RNAi efficiency is shown only for day 5 and 7. For the differences to be attributed to the RNAi, the authors should show that the RNAi is already active as early as day 2.

In figure 6C, ds-GFP males are mated with wildtype females and the average egg count is equal to 7-8, while ds-GFP female from 6D lay in average 2.5 eggs. This would suggest that dsRNA injection itself has a dramatic effect on egg count, effect larger than the nature of the dsRNA itself. This should at least be discussed.

Once again, no information is provided for this figure on the statistical tests used to interpret the data.

  • On the discussions:

The authors raise several interesting discussion points.

For example, lines 364-375, the authors discuss that the differential expression of the gene timeless could indicate an involvement of the circadian cycle in the mating process. To make this hypothesis more relevant, the authors should clearly state in the Materials and Methods section if all the samples used for RNAseq and RT-qPCR were strictly collected at the same time of the day.

The authors insist on the potential importance of the gene yellow g. However the comparison with yellow g in D. melanogaster appears irrelevant as no sex combs are present in B. olea. The authors focused their study on the reproductive tissues, therefore hypotheses about the function of yellow g and any other genes outside these tissues cannot be done based on this research. Such hypotheses should be removed from the text.

Minor comments:

Lines 253-286: The whole paragraph is not supported by any figure, except a figure S1 I could not access during the review process. The text is therefore rather difficult to follow. I would suggest the authors to provide visual support for this paragraph.

On the transition from the RNAseq analysis (part 3.1 and 3.2) to the expression profile analysis (3.3), the logic behind why these genes were chosen and not other is not clear. Especially some of the genes with unknown function are not discussed further and are not used for functional analysis. Could the authors provide the readers with more details about these genes and why they carried on studying them?

Reviewer 2 Report

This is a well written presentation of a relatively comprehensive investigation into the levels of transcription of Olive fly genes in both males and females pre- and post-copulation. I have only a few suggestions that I believe will improve the manuscript:

  1. Table 1 should be reconfigured to eliminate the distracting numbers on the left and generally improve the presentation of these data.
  2. Please, edit the caption of Table 1 to provide clarity. For example, you might change "For genes expressed in the tissue of testes positive value of logFC represents the over expression.." to "A positive value of logFC indicates over expression ..."
  3. Line 325, the word "overexpression" is incorrect in this context. Use instead, "increased expression", "greater expression", "relatively greater expression" or "increased transcript abundance".
  4. Please, more explicitly describe the preparation of biological and experimental replicates in the Materials and Methods.
  5. Based on the text, the y-axes of Figures 6A and 6B are incorrect. The figures show proportions, not percentages.

Round 2

Reviewer 1 Report

In the new version of the manuscript, the authors answered most of the concerns raised during the first review.

The description of the experimental design and the results have gained both in content and clarity and the statistical analysis are clearly indicated for all experiments. Also all RT-qPCR experiments were reanalyzed with an additional reference genes, making the data more robust. To my point of view, the discussion stays somehow over-interpreted, as the functional study remains exploratory.

However, I still have concerns about the availability of the RNAseq data presented here. If data obtained with Ion Proton are easy to access via GEO database, it is still not clear were testes data can be found, who generated them, and if they are published already.

Regarding the testes from mated males:

The authors indicate that the sample from testes of mated males can be found in SRA database PRJNA288990. If I am correct, it is the sample with accession number SRX5589539. Could the authors mention it more clearly, as it nowhere indicates that the sample originate from mated males?

Also the data contained in PRJNA288990 are already associated with a publication by Bayega et al., 2020 in BMC Genomics (https://doi.org/10.1186/s12864-020-6672-3) and this is not mentioned anywhere in the text. The authors need to clearly state that this data is already published and clearly reference the corresponding paper.

Regarding the testes from virgin males:

The data regarding the testes from virgin males are impossible to find. The paper in which these data are published (Sagri et al., 2014) does not contain any accession number I could fine. The authors should clearly provide the reader with an accession number for these data. If the data were unfortunately not deposited in a database, it would be the perfect opportunity to do it now, as it seems they were generated by the same lab.

Number of replicates per experiment:

With all this taken into account, I found out that only one replicate is available per treatment for testes samples, whereas all other samples were obtained in duplicates. Analysis of RNAseq data without replicates is much less powerful and mainly exploratory. The authors should clearly mention in the text when data do not have replicates. It is currently not stated clearly enough.

Supplementary Figure S6

Finally, the new supplementary figure S6 provides the reader with useful information on the sequencing libraries. However testes samples are nowhere mentioned in the table 6.1. Even if the data were generated for different publications, the authors have probably reanalysed them with their new transcriptome assembly and therefore should indicate how the new assembly performed with these samples.

Or maybe they reuse the assembly from Bayega et al., 2020, which seems to be based on the same Illumina  data and using the same Trinity based pipeline. This point also is not clear and should be clarified.

Also MAGSB1 and FAGsB2 do not cluster with their corresponding replicates in the PCA plot. This should be at least mentioned in the text, if not explained. The legend should also be more extensive, clearly stating which samples are from virgin or mated flies (A or B), and if possible use the same names as in the main text.
